# Pathophysiology and Neuroimmune Interactions Underlying Parkinson’s Disease and Traumatic Brain Injury

**DOI:** 10.3390/ijms24087186

**Published:** 2023-04-13

**Authors:** Alyssa Lillian, Wanhong Zuo, Linda Laham, Sabine Hilfiker, Jiang-Hong Ye

**Affiliations:** 1New Jersey Medical School, Rutgers University, 185 South Orange Avenue, Newark, NJ 08901, USA; 2Department of Anesthesiology, Pharmacology, Physiology & Neuroscience, New Jersey Medical School, Rutgers University, 185 South Orange Avenue, Newark, NJ 08901, USA

**Keywords:** α-synuclein, microglia, neuroinflammation, neurodegeneration, neuromelanin, oxidative stress, Parkinson’s disease, S100A9, S100B, TBI

## Abstract

Parkinson’s disease (PD) is a progressive neurodegenerative disorder clinically defined by motor instability, bradykinesia, and resting tremors. The clinical symptomatology is seen alongside pathologic changes, most notably the loss of dopaminergic neurons in the substantia nigra pars compacta (SNpc) and the accumulation of α-synuclein and neuromelanin aggregates throughout numerous neural circuits. Traumatic brain injury (TBI) has been implicated as a risk factor for developing various neurodegenerative diseases, with the most compelling argument for the development of PD. Dopaminergic abnormalities, the accumulation of α-synuclein, and disruptions in neural homeostatic mechanisms, including but not limited to the release of pro-inflammatory mediators and the production of reactive oxygen species (ROS), are all present following TBI and are closely related to the pathologic changes seen in PD. Neuronal iron accumulation is discernable in degenerative and injured brain states, as is aquaporin-4 (APQ4). APQ4 is an essential mediator of synaptic plasticity in PD and regulates edematous states in the brain after TBI. Whether the cellular and parenchymal changes seen post-TBI directly cause neurodegenerative diseases such as PD is a point of considerable interest and debate; this review explores the vast array of neuroimmunological interactions and subsequent analogous changes that occur in TBI and PD. There is significant interest in exploring the validity of the relationship between TBI and PD, which is a focus of this review.

## 1. Introduction

Parkinson’s disease (PD) is a progressive neurodegenerative disorder that affects approximately 0.3% of the general population, making it the second most common neurodegenerative disease [1,2]. In the US, about 60,000 people are diagnosed with PD yearly [3]. It is characterized by a set of motor symptoms, including postural instability, bradykinesia, resting tremors, and dysfunctional gait, which are typically accompanied by autonomic and psychiatric symptoms such as constipation, urinary urgency, orthostatic hypotension, hypersalivation, forgetfulness, depression, and sleep disorders [2,4,5,6]. The diagnosis of PD is usually based on a medical history, a review of signs and symptoms, and a neurological and physical examination by a trained neurologist. PD is progressive, and its symptoms worsen over time. There are five stages of PD, and clinicians use them to describe how motor symptoms progress in PD. There is currently no known cure for PD, but research is ongoing to develop new treatments and improve the care of people living with the disease.

Evidence has shown that one of the risk factors implicated in the pathogenesis of PD is prior traumatic brain injury (TBI): some studies cite a link between TBI and the development of PD [5,7,8,9,10,11,12,13,14,15,16,17]. Clinical studies have also shown that previously concussed athletes are at higher risk of developing PD-like syndrome [18,19]. According to the Centers for Disease Control and Prevention (CDC), there were approximately 223,135 TBI-related hospitalizations in 2019, and 64,362 TBI-related deaths in 2020 [20]. Overall, mild TBI is the most common TBI sustained [21].

This paper will highlight and explain the shared pathomechanistic link between TBI and PD and shed light on the multiple clinical implications of these two related conditions. Here, we will examine the pathophysiology of PD, including the roles of α-synuclein, neuromelanin, iron, S100A9, S100B, cytokines, and inflammatory pathways driven by microglial activation. We will also discuss current pharmacologic and non-pharmacologic treatments alongside future treatment options, the development of which depends on a more in-depth understanding of the etiology and mechanisms underlying PD. Furthermore, we will describe the mechanisms of synaptic plasticity in PD and analyze several of the fundamental and analogous aspects underlying the pathophysiology of both TBI and PD. Neuroinflammation induced by TBI promotes a pro-inflammatory state characterized by reactive oxygen species (ROS) and oxidative stress, and is influenced by aquaporin-4 (AQP4) [22]. This pro-inflammatory state leads to dopaminergic abnormalities, α-synuclein aggregation, S100A9 and iron accumulation, mitochondrial dysfunction, and glutamatergic excitotoxicity [23]. In a genetically susceptible individual, these alterations caused by TBI, which mimic the pathologic changes present in idiopathic PD, are likely sufficient to signal the development of full-blown PD.

## 2. Pathophysiology of PD

### 2.1. Etiology and Mechanisms of Neuronal Pathology

The clinical presentation of PD is accompanied by a loss of A9 dopaminergic neurons in the substantia nigra pars compacta (SNpc) and the accumulation of α-synuclein containing Lewy bodies and Lewy neurites [2,4,5,24]. Lewy body formation is not limited to the SNpc; it has been seen in serotonergic, cholinergic, and noradrenergic circuits throughout the brain and various structures of the autonomic nervous system [2]. As such, it is not only dopamine levels that are affected in PD, but serotonin, acetylcholine, and norepinephrine as well [25]. The A9 neurons represent the ventrolateral group of cells and appear to be the most susceptible cells within the SNpc, perhaps due to their intrinsic pacemaker properties, necessitating repeated calcium influx through L-type calcium channels. However, these cells may have insufficient mechanisms to stabilize the concentration of free calcium ions, leading to increased stress and homeostatic imbalance over time [6]. α-synuclein pathologically self-assembles into amyloid oligomers, forming aggregates that, alongside mitochondrial dysfunction, may lead to the loss of striatal dopaminergic neurons [2,26]. The loss of these dopaminergic neurons has been shown to correlate with a significant decrease of tyrosine hydroxylase expression in SNpc neurons, the rate-limiting enzyme of catecholamine biosynthesis [27]. Furthermore, neuronal loss in the SNpc directly parallels the dopamine transporter’s transcription level [2]. Notably, striatal dopaminergic neuron loss must exceed 60% before symptoms of PD become clinically apparent [2,4,28]. This may be due to intrinsic compensatory mechanisms that maintain tonic dopamine release, thereby functionally mimicking a standard and intact nigrostriatal dopaminergic system [29].

Although the etiology of PD is complex and not fully understood, key components which may contribute to disease progression have been identified, such as α-synuclein accumulation [15,26], mitochondrial dysfunction, oxidative and endoplasmic reticulum stress [2,4,24,30,31], MHC (major histocompatibility complex) class II driven neuroinflammation [5,24,26], glutamatergic excitotoxicity [1], neuromelanin and iron deposition [32,33,34], and genetic and epigenetic changes [4,35]. These factors, which all play essential roles in the pathogenesis of PD, promote protein misfolding and aggregation, which ultimately leads to neuronal death. It is likely that a combination of genetics and environmental influences underlie these pathologic changes. The exposure to pesticides, herbicides, and heavy metals increases the risk of developing PD [25]. Familial forms of PD have been linked to mutations in numerous genes, including but not limited to SNCA, PARK2, PINK1, PARK7, LRRK2, BST1, MAPT, and VPS35 [4,6,24]. Rare familial mutations in the α-synuclein gene (SNCA) lead to an early onset and aggressive form of PD [36,37,38,39]. The Braak hypothesis proposes an alternative pathogenesis of PD, similar to that of a prion-like disease [40]. The proposed mechanism states that the condition begins sporadically following pathogen entry and eventually leads to the neurodegenerative changes that underlie PD. In this disease model, PD starts in the medulla and olfactory bulb and later ascends to the SNpc, causing pathologic changes that result in symptom presentation [2,40,41,42].

### 2.2. α-synuclein

α-synuclein is a small 140 amino acid presynaptic protein that functions in synaptic plasticity and vesicle packaging, trafficking, and release [6,24,27,43,44]. Specifically, it is believed to regulate the number of synaptic vesicles docked at the synapse during neurotransmitter release [45]. α-synuclein is typically unfolded and un-phosphorylated; its presence as an oligomer in cytoplasmic inclusions suggests a pathologic process. Hyperphosphorylation at the serine 129 residue of α-synuclein [2,5] is commonly seen in the brains of PD patients, with only 4% of α-synuclein phosphorylated in the brains of individuals without PD versus 90% in PD brains containing Lewy bodies [24]. In its folded and aggregated state, α-synuclein impairs subcellular processes and enhances the formation of Lewy bodies, further contributing to PD progression [2,4,24]. For example, α-synuclein aggregates can cause mitochondrial dysfunction by impairing fundamental mechanisms needed for ATP production, creating an uninhabitable environment for the high-energy A9 dopaminergic neurons [5]. Notably, α-synuclein has been reported to generate oxidative stress directly by producing a superoxide and indirectly by activating microglial cells through binding to the TLR2, TLR4, and CD11b receptors which initiate nuclear factor kappa-B (NF-ĸB) and mitogen-activated protein kinase inflammatory pathways [46,47,48]. However, this has not been independently validated.

### 2.3. Neuromelanin and Iron

Neuromelanin is a dark brown-colored molecule containing high levels of iron found within intracellular granules. Neuromelanin is first evident in the SNpc when a child is about three years old and naturally accumulates with age, serving as a risk factor for developing PD [33,34]. In addition to the SNpc, levels of iron increase with age in microglia and astrocytes throughout the cortex, basal ganglia, cerebellum, amygdala, and hippocampus [34]. In the early stages of PD, α-synuclein becomes trapped within neuromelanin granules, forming aggregates. There is preferential inflammation and degeneration of neuromelanin neurons alongside the accumulation of α-synuclein in Lewy bodies. A recent rodent model overexpressing tyrosinase, a rate-limiting enzyme that is involved in the synthesis of melanin in the substantia nigra was developed to assess the consequences of neuromelanin accumulation. In rodents, neuromelanin levels paralleled those seen in elderly human brains, suggesting that tyrosinase may contribute to neuromelanin development [49]. The rodents developed a PD-like pathology when intracellular neuromelanin accumulation reached a certain threshold. It appears that an increase in intracellular neuromelanin levels precedes the loss of dopaminergic neurons and therefore plays a pivotal role in the neuronal degeneration seen in PD [33]. Excessive cytosolic or extraneuronal dopamine can give rise to nonselective protein modifications. The excessive formation of neuromelanin from the oxidation of dopamine may additionally promote pathologic changes leading to the development of PD [50], since excessive neuromelanin accumulation interferes with typical proteostasis cellular systems [51]

Neuromelanin may also promote oxidative stress and create a toxic environment within the brain due to its ability to chelate and form complexes with iron, as iron directly contributes to ROS generation, oxidative stress, and protein aggregation. Neuromelanin-iron complexes and the consequences of iron chelation are evident in an increase in redox-active iron in SNpc neurons. The increase in redox-active iron is more pronounced in PD patients with substantial neuronal loss, although their neuronal loss is not necessarily limited to the SNpc [34]. Within the dopaminergic neurons of the SNpc, there is an altered expression of DMT1 (divalent metal transporter 1), a divalent cation transporter involved in the movement of iron across cellular membranes [52]. In addition to the changes seen in iron transport, there is an increase in ferritin, the principal iron storage protein, in PD [32,34]. Neuromelanin is the primary iron storage molecule within SNpc dopaminergic neurons [33], and there is potential for neuromelanin to be a source of toxic iron if its binding capacity is saturated. In general, there tends to be increased iron levels in extrapyramidal brain regions at baseline, perhaps owing to the finding of iron imbalance associated with movement disorders such as PD [53]. This is further evidenced by the increased iron concentrations in the SNpc of rodents treated with methyl-4-phenyl-1,2,3,6-tetrahydropyridine (MPTP), which induces dopaminergic neuron loss and is used to reproduce PD motor symptoms in rodent models [34,54]. Similarly, the intraventricular administration of deferoxamine, an iron chelator, has been reported to be protective against treatment effects with MPTP in mice [54].

A few papers have suggested that AQP4 may be necessary for regulating dopamine levels in the brain: AQP4 knockout mice were shown to have increased extracellular levels of dopamine at baseline [55]. Furthermore, AQP4 is expressed in adult neural stem cells and is essential in neurogenesis [56]. Secondary to its actions within dopaminergic and neuronal systems, AQP4 has been implicated in the pathogenesis of underlying PD. Knockout AQP4 mice treated with MPTP had a more pronounced inflammatory response with increased activation of the NF-ĸB pathway and more significant dopaminergic neuron loss compared to wild-type mice [57,58]. The regulation of the inflammatory response in PD by AQP4 is supported by its expression throughout the immune system, specifically in mouse thymus, spleen, and lymph nodes, and further detected in B cells, T cells, macrophages, and natural killer cells taken from these immune organs [57]. Additional support comes from studies showing decreased blood levels of AQP4 in patients with PD compared to controls, but further research is needed to confirm the role of AQP4 in PD pathogenesis [59].

### 2.4. Microglia

Microglia, the resident macrophages in the brain, generate superoxide anions from nicotinamide adenine dinucleotide phosphate oxidase (NOX2) for their anti-microbial actions. NOX2, a phagocytic NADPH oxidase, produces ROS from the respiratory burst during phagocytosis and forms a superoxide by transferring electrons from NADPH to reduce oxygen [60]. This ROS-mediated inflammation is exacerbated by mutated forms of α-synuclein in the microglia, which can cause greater NOX2 activation in the microglia than their wild-type counterparts, resulting in elevated ROS production [61]. Although microglia help clear debris from apoptotic dopaminergic neurons, they are also potent mediators of neuroinflammation. Their induced production of chemokines may be deleterious to surviving dopaminergic neurons already exhibiting mitochondrial dysfunction [24]. By these mechanisms, microglia contribute to oxidative stress by stimulating oxidative bursts and pro-inflammatory cytokines [1,62].

### 2.5. S100A9

S100A9 is a calcium-binding protein produced by neuronal and microglial cells that belongs to a larger family of proteins involved in numerous neurodegenerative, inflammatory, and malignant conditions [26]. Although there have been few studies on the role of S100A9 in PD, one study showed that S100A9 and α-synuclein co-aggregate in pathologic states. In PD brain tissue, S100A9 and α-synuclein were noted to be present in 77% of neuronal cells in the substantia nigra, and in 20% of Lewy bodies within the substantia nigra and frontal lobe [26]. Chronic neuroinflammation promotes the spread of S100A9 in brain tissues. It may initiate the co-aggregation of S100A9 and α-synuclein that underlies the pathogenesis of PD, though further research is needed to validate a relationship between S100A9 and PD [26].

### 2.6. S100B

In addition to S100A9, studies have shown that S100B, another calcium-binding protein in the same family, is implicated in neurodegenerative diseases, such as PD [26,63,64,65]. S100B is predominantly expressed in astrocytes but also in other cell types, such as neural progenitor cells, dendritic cells, and mature oligodendrocytes [65]. S100B acts as a damage-associated molecular pattern (DAMP), and is therefore considered a marker of brain damage when present. S100B signals ROS production and pro-inflammatory cytokines at micromolar levels, promoting intracellular calcium overload, apoptosis, oxidative damage, and neuroinflammation. Current evidence suggests a time-specific role of S100B in PD, with higher levels seen in earlier disease states and normalized levels seen years later [65]: serum autoantibodies against S100B were shown to be four times higher in PD patients five years post-diagnosis, with antibody levels decreasing to typical baseline values ten years after diagnosis [30]. Even so, human post-mortem studies have found elevated levels of S100B in the SNpc and dorsomedial prefrontal cortex of PD patients [63,65].

### 2.7. Synaptic and Cellular Plasticity

It has been posited that there is increased dopamine synthesis in the remaining dopaminergic neurons and the preferential release of dopamine to damaged areas, which allows for preserving dopamine delivery to striatal regions [29]. The loss of dopaminergic neurons prompts cellular and synaptic plasticity within the striatum, leading to hyperactivity of spiny projection neurons within the indirect pathway of the basal ganglia, the inhibition of external globus pallidus neurons, and the disinhibition of the subthalamic nucleus, which results in the inhibition of brainstem motor regions [66]. Homeostatic plasticity can be seen in direct pathway spiny projection neurons; their increase in intrinsic excitability aims to balance the concurrent loss of dopaminergic neurons [66]. Synaptic plasticity within the central nervous system is also driven by AQP4, a bidirectional water channel protein found in the blood-brain barrier (BBB) and cerebrospinal fluid barriers. Aside from its role in water movement, AQP4 is essential in synaptic plasticity and astrocyte migration [58]. Astrocytes and ependymal cells highly express AQP4; the former provides structural and functional support for the brain, is an essential regulator of potassium homeostasis, and can influence synapse formation [58].

## 3. Treatment Modalities

### 3.1. Pharmacologic Treatments

Knowledge of the pathophysiologic mechanisms underlying PD has been crucial to developing more precise and effective treatment options. Pharmacologic treatments for PD primarily aim to reestablish normal dopamine levels, and include medications such as carbidopa-levodopa, dopamine agonists, anticholinergic agents, catechol-O-methyltransferase inhibitors, and monoamine oxidase-B (MAO-B) inhibitors (Table 1). While helpful in alleviating many symptoms of PD, these medications are not without side effects. Levodopa provides the most significant relief of motor symptoms but risks dose-dependent dyskinesia [25]. Dopamine agonists and MAO-B inhibitors carry less of a risk of dyskinesia, but at the cost of lesser symptom improvement. Anticholinergic agents are helpful for those with significant tremors but can cause adverse cognitive effects [25]. Many individuals treated with oral dopamine agonists experience impulse control disorders such as gambling and compulsive spending [67]. Over time, individuals tend to require more frequent and significant Levodopa dosages to control symptoms. Interestingly, this is not due to tolerance but to loss of response to dopamine and dopaminergic medications secondary to pathologic neuronal changes [68]. BBB-permeable iron chelators such as VK-28, HLA-20, and M30 can offer therapeutic benefits, though more data is needed to determine the efficacy of such treatments [34,69].

It is important to note that this table is not exhaustive, and there may be other drugs used to treat Parkinson’s disease. Some medications may have multiple brand names and may be used for other conditions.

A recent phase two trial evaluated nilotinib, a protein tyrosine kinase inhibitor that targets BCR-ABL, c-kit, and PDGF as a potential disease-modifying drug in PD [73]. Several studies using animal models of synucleinopathies have found that nilotinib can enter the brain and degrade α-synuclein and Tau proteins, which may account for its utility as a potential therapy for PD. However, it has not yet been studied to determine whether it could benefit TBI patients [75,76,77,78]. Nilotinib was deemed to be safe and well-tolerated in the phase two trial, and a phase three trial is likely to be developed moving forward [73]. Additionally, immunization therapies are being developed that target distinct α-synuclein epitopes. Preclinical immunotherapy studies in PD mouse models show reduced α-synuclein aggregates and improved motor and cognitive function [79]. Currently, multiple phase-two clinical trials are evaluating the effectiveness of Prasinezumab, a humanized monoclonal antibody directed against aggregated a-synuclein, in patients with early PD [74]. Ongoing phase one and two clinical trials utilize immunotherapies targeting α-synuclein; however, preliminary data has not shown significant disease amelioration thus far in TBI patients [79].

### 3.2. Non-Pharmacologic Treatments

Non-pharmacologic treatments with proven benefits include exercise regimens such as gait, balance, resistance, and strength training. Speech, physical, and occupational therapies also help maintain or improve motor and autonomic function [25]. Similarly, exercise therapy in patients with mild to moderate impairments in gait and balance post-TBI has been shown to improve both static and dynamic gait and postural stability [80]. Deep brain stimulation is an invasive treatment option for those whose symptoms are not adequately controlled with medication alone. It involves the surgical placement of wires in the subthalamic nucleus or globus pallidus interna that are connected to a battery implanted in the chest wall. Following implantation, personalized stimulation parameters are set to provide optimal control over motor symptoms, tremors, and dyskinesia [25]. Future therapies are likely to expand the realm of non-pharmacologic treatment by focusing on novel therapeutics, including but not limited to the grafting of dopaminergic neurons, stem cell therapy, glial-derived neurotrophic factor (GDNF) therapy, and gene therapy using adeno-associated virus (AAV) vectors [81].

PD is a neurodegenerative disorder characterized by both motor and non-motor symptoms. While pharmacological treatments are commonly used to manage the condition, non-pharmacological therapies have also shown promise in improving symptoms. Table 2 outlines some of the non-pharmacological treatments for PD and their respective mechanisms.

A multidisciplinary approach incorporating pharmacological and non-pharmacological treatments may be the most effective way to manage PD. However, these treatments are believed to enhance the function of the dopamine-producing neurons in the brain, reduce inflammation, and improve overall brain health [85]. It is important to note that the mechanisms by which non-pharmacological treatments improve symptoms in PD are yet to be fully understood.

## 4. Neuroimmune and Inflammatory Responses to TBI

### 4.1. Cellular and Inflammatory Changes

Minutes after a TBI occurs, reactive oxygen species (ROS) and purines are released, initiating an inflammatory cascade [86]. Within 24 h after brain injury, there are increases in IL-1β, IL-6, IL-10, and TNFα, with levels peaking between 3–9 h post-injury [22,87,88]. Neutrophils are rapidly mobilized to the CNS, entering the choroid plexus and meningeal blood vessels. In the CNS, they recruit monocytes, release metalloproteinases, proteases, and TNFα, and generate ROS. Within the first week of sustaining a TBI, cell necrosis occurs secondary to direct trauma to brain tissue and neuronal death from inflammatory changes and axonal pathology induced by ROS [86]. Inflammation decreases between one week and one month post-TBI, which is when and neuronal remodeling is seen. There is an increase in chronic pathologies, such as neurodegeneration, evidenced by the misfolding of the α-synuclein, APP, Tau, and TDP43 proteins [5]. Although this altered pathology is not specific to PD, the protein misfolding seen may indicate a general deficit in proteostasis (for a review on this topic, see ref. [4]). Notably, TDP43 has been found in the brains of humans with chronic traumatic encephalopathy (CTE), PD, and dementia with Lewy bodies [89], as well as in rats [90]. From one month to years after sustaining a TBI, there is wide variability in the neural changes patients experience, with some reaching full recovery with minimal lasting damage. In contrast, other patients experience lifelong neurodegenerative changes that can lead to cognitive decline and the eventual development of disease (Figure 1).

Aquaporins (AQPs) are a family of transmembrane proteins that facilitate the movement of water and other small solutes across cell membranes [91]. AQPs are present in various tissues and are involved in physiological processes such as fluid homeostasis, urine concentration, and cell migration [92]. One specific AQP of interest is aquaporin-4 (AQP4), which is mainly expressed in astrocytic foot processes in the central nervous system (CNS). AQP4 has been found to play a role in regulating water homeostasis in the CNS, and is also involved in the pathology of certain CNS disorders. Recent research has suggested that some AQPs, including AQP4, may also facilitate the diffusion of hydrogen peroxide (H2O2) and other gas molecules across the cell membranes [93,94]. These AQP-mediated gas conduction pathways are not yet fully understood, and their physiological relevance is still under investigation [94]. Furthermore, AQPs, including AQP4, have been implicated in the permeabilization of the blood-brain barrier (BBB) in cerebral ischemia/reperfusion injury (CIRI) [95]. This suggests that AQPs may play a role in the development and progression of neurological disorders. It is also worth noting that the dysregulation of the AQP function can lead to oxidative stress and other pathological processes [92]. AQP-mediated pathways have been identified in cell membranes’ lipid and protein domains. They may contribute to water movement across cellular plasma membranes by a diffusion or line-alanine mechanism [91,96]. The two concepts are increased AQP expression and redistribution/surface localization. For example, previous studies have shown an increase in AQP4 membrane localization in primary human astrocytes, which was not accompanied by a change in AQP4 protein expression levels. This mislocalization can be a potential therapeutic target [97,98,99].

Remarkably, Kitchen et al. [100] demonstrated that the development of edema following injury-induced hypoxia is AQP4 dependent. Their study showed that CNS edema is associated with increased total aquaporin-4 expression and aquaporin-4 subcellular translocation to the blood-spinal cord-barrier (BSCB). The pharmacological inhibition of AQP-4 translocation to the BSCB prevents the development of CNS edema and promotes functional recovery in injured rats. This role has recently been confirmed by Sylvain et al. [101], demonstrating that targeting AQP4 effectively reduces cerebral edema during the early acute phase of stroke using a photothrombotic stroke model. They have also shown a link to brain energy metabolism, as indicated by the increase of glycogen levels. In addition to modulating the inflammatory response, AQP4 also impacts inflammatory processes following TBI. Mouse models show that knockout AQP4 mice had decreased intracranial pressure, water accumulation, and cell death, along with smaller lesion volumes directly following brain impact, compared to wild-type mice [58]. Knockout mice also showed reduced astrogliosis, a pathologic process often seen following neuronal destruction from various causes such as trauma, infection, ischemia, or neurodegenerative disease [58]. AQP4 contributes to edematous states differently depending on the length of time post-injury. During the primary phase following TBI, which is characterized by vasogenic edema, AQP4 is downregulated. Later, AQP4 is upregulated during the second phase, commonly characterized by cellular edema. A lack of AQP4 during the primary phase would decrease the amount of water able to leave the brain and promote an edematous state. In contrast, downregulation during the second phase would have the opposite effect [58,102]. Studies of various TBI models have shown that the maximal downregulation of AQP4 occurs at one-day post-TBI when the brain is most edematous, and this downregulation persists for about 48 h. The downregulation of AQP4 was most prominent in locations where BBB disruption was most significant, and AQP4 was relatively spared where BBB damage was minimal [58]. Taken together, this information suggests a protective role for AQP4 in edematous states occurring post-TBI, likely via extracellular water removal, which reduces increased intracranial pressure and edema formation.

AQPs have been validated as an essential drug target, but no single drug has yet been approved to target it successfully; drug discovery for AQPs has made little progress due to a lack of reproducible high-throughput assays and difficulties with the druggability of AQP proteins. However, recent studies have suggested that targeting the trafficking of AQP proteins to the plasma membrane is a viable alternative to the direct inhibition of the water-conducting pore. A recent article reviewed the literature on the trafficking of mammalian AQPs to highlight potential new drug targets for various conditions associated with disrupted water and solute homeostasis [103,104].

### 4.2. BBB Integrity, Cytokines, and Chemokines

TBI causes a breakdown of the BBB, induces pro-inflammatory reactions in the brain, and leads to secondary brain injury characterized by disrupted mitochondrial function, the perturbation of calcium homeostasis, and glutamate excitotoxicity [1,11,21,105,106]. The tight junctions of the endothelial cells that line the BBB are disrupted by inflammatory mediators, allowing for the entry of iron, ROS, toxins, and neutrophils, all of which augment cytokines and inflammatory levels. The loss of BBB integrity promotes neural edema, hemorrhage, hypoxia, and vascular leakage, leading to cell death in brain parenchyma. Neuronal death can also occur from mechanical forces—shearing injuries can cause the tearing of axonal fibers and disrupt gray and white matter [86]. Mechanical shear stress can activate the transient receptor potential cation channels (TRP channels) TRPV1 and TRPA1. Activating these channels leads to the release of various neuropeptides, including substance P. Substance P activates the microglia and astrocytes, promotes leukocyte migration, and can cause the degranulation of mast cells, all of which alter the permeability of the BBB [22].

The release of DAMPs such as DNA, RNA, heat shock proteins, and HMGB1 can lead to microglial activation, which, if persistent, contributes to secondary brain injury and neurodegeneration [22,86,105,106]. M1 microglia promote a pro-inflammatory state, whereas M2 microglia suppress the inflammatory response and promote tissue remodeling. Inappropriate or prolonged M1 activation can cause persistent tissue damage years after the initial injury [22]. Similarly, CNS and peripheral immune cells respond to the injury and participate in repair, but can also promote secondary injury by releasing inflammatory cytokines and chemokines; such acute cellular reactions are mediated by astrocytes, microglia, macrophages, neutrophils, and T cells [86,87]. Immune cells are additionally triggered by purinergic receptor signaling. These transmembrane receptors detect ATP, ADP, and adenosine; the release of ATP from damaged cells activates purinergic receptors that further drive TBI pathogenesis. Microglial responses depend on the purinergic receptors P2 × 4, P2Y_6_, and P2Y_12_, whereas P2 × 7R signaling is necessary for neutrophil recruitment [86]. TBI can induce inflammatory and immunological cascades through these mechanisms, promoting cellular and neuronal damage.

## 5. Relationship between TBI and PD

### 5.1. Oxidative Stress

While there is currently no consensus in the literature regarding the mechanistic relationship between TBI and the development of PD, and further correlational evidence is warranted, there are noticeable pathologic changes in the brain following a TBI, many of which mimic the pathology of PD (Figure 2). ROS and elevated levels of α-synuclein produced following TBI promote a pro-inflammatory cascade, which can create an environment conducive to the development of PD [5,27]. S100A9, a significant marker of brain injury, also drives the pro-inflammatory cascade. Shortly after a TBI, S100A9 is present in human brain tissue as amyloid oligomers [15,107]^,^ and has been seen in mice brain tissue on days 1 and 3 following controlled cortical impact TBI, with a sharp decrease in S100A9 levels seen in 70% of the mice by day 10 [15]. Moreover, the expression of the dopaminergic markers tyrosine hydroxylase and dopamine transporter (DAT) were downregulated for as long as 30 days post-TBI [108].

The current literature points to the abnormal regulation of iron in the injured brain. Oxidative stress in the brain is furthered by the consequences of iron deposition and its participation in Fenton and Harbor-Weiss reactions. These reactions are characterized by the generation of hydroxyl and superoxide radicals from the interaction of ferrous (Fe^2+^) and ferric (Fe^3+^) iron with hydrogen peroxide and oxygen. Fe^2+^ and Fe^3+^ can also react with lipids, catalyzing the formation of peroxyl and alkoxyl radicals. Free iron deposition is associated with heme degradation, and is the more damaging form of iron, as it is a potent source of oxidative stress. On the other hand, iron deposition in its heme-bound form is associated with intracranial hemorrhage, a common feature of TBI [32]. The importance of iron for optimal mitochondrial function and myelin generation by oligodendrocytes, results in significant harmful effects such as protein aggregation, lipid peroxidation, and amyloid deposition [32,109]. TNFα and IL-6 encourage iron accumulation in both neurons and microglia and induce the expression of iron transport receptors [109]. As with PD, iron accumulation is seen in various neuronal structures following TBI, where it promotes neurodegeneration [32,34].

### 5.2. α-synuclein and Dopamine

α-synuclein aggregation has been implicated as a direct response to TBI, as elevated levels of α-synuclein are seen in the CSF and brains of patients following TBI [108,110], with levels positively correlated with the severity of injury sustained [21,111,112] and inversely correlated with the chance of survival [28,111]. Three-fold increased α-synuclein accumulation is present in the ipsilateral SNpc in TBI animal models as compared to the contralateral SNpc or controls [27], and the elevated expression of α-synuclein is still discernable 60 days post-TBI [27,43]. These results are consistent with the high propensity of α-synuclein to misfold and aggregate as a response to post-translational modifications, oxidative stress, mutations, and environmental changes [108]. While the exact role of post-translational modifications and other epigenetic changes following TBI has not been fully elucidated, a repeated blast injury model of TBI in rats showed DNA methylation in neurons and glia within the TGF-beta pathway, with the degree of injury sustained likely influencing the amount of methylation seen [35,113].

Dopamine abnormalities are a common finding following TBI, in part due to the high susceptibility of dopaminergic neurons to the inflammation and oxidative damage mediated by microglia [108]. Patients with early PD and moderate to severe TBI exhibit a similar reduction in DAT binding within the caudate [114]. Animal models of TBI show a loss of dopaminergic fibers and an associated hypo-dopaminergic state post-TBI [115]. Experimentally induced TBI in rats was shown to cause a 15% loss of dopaminergic neurons ipsilateral to the injury at 11 days post-TBI, which increased to 30% bilateral dopaminergic neuron loss at 26 weeks post-TBI [116,117]. Furthermore, at 60 days post-TBI, there were decreased markers of dopamine synthesis [27,28].

### 5.3. Clinical Relevance

A study of adults aged 55 and up, without any form of PD or other neurodegenerative conditions, who presented to the emergency department after a TBI, found that they had a 44% increased risk of developing PD over the next five to seven years [28]. Overall, there was evidence that patients who have sustained a previous TBI are more likely to be diagnosed with PD later than individuals with no prior history of TBI. This study demonstrated that various risk factors contribute to the development of PD [28], and that TBI may be such a risk factor, increasing the likelihood that a susceptible individual will develop PD [110,114,118]. It has been hypothesized that TBI-produced brain injury could reduce the motor reserve and lead to a vulnerable patient’s earlier diagnosis of PD. It is also possible that TBI may advance or alter a pre-existing neurodegeneration cascade [28]; for example, variability in the gene encoding α-synuclein has been associated with an increased risk of developing PD after TBI, such that the increased levels of α-synuclein seen post-TBI could further the development of PD in someone with said genetic variability [114,118,119,120]. Similarly, specific single nucleotide polymorphisms associated with mitochondrial haplogroups H, U, V, W, X, J, Y, but not K or T, have been deemed risk factors for neurodegenerative conditions such as PD. This has potential clinical implications given the significant degree of mitochondrial dysfunction seen post-TBI, as only patients with haplogroup K were shown to have a better neurologic outcome [120].

## 6. Conclusions

Parkinson’s disease is a disorder of complex etiology, with associated characteristic changes that lead to subsequent neurodegeneration and clinical symptoms. One clinical risk factor that has been suggested as causative for PD is prior TBI. The most plausible and fitting mechanism linking TBI to PD proposes that previous TBI promotes neurodegeneration and the development of PD in susceptible individuals by creating a neuroimmunological environment similar to that of primary PD. Sufficient evidence outlines the pathological changes following TBI and their similarity to dopaminergic abnormalities, α-synuclein aggregation, S100A9 accumulation, and pro-inflammatory cytokines and ROS release in PD patients. Oxidative stress is a crucial mechanism underlying neuroinflammation, mitochondrial dysfunction, and glutamatergic excitotoxicity in TBI and PD. AQP4 influences the inflammatory response in TBI and PD. Additionally, it contributes to synaptic plasticity, astrocyte migration, and neurogenesis in PD. Increased neuronal iron accumulation is evident in neurodegenerative states such as PD and post-TBI. However, it remains to be seen whether iron accumulation occurs secondary to neuronal loss or the BBB’s increased permeability allows for greater iron entry into the brain. While there are currently no preventive or specific treatments for PD following a TBI, a neurology referral after sustaining a TBI may allow for the early identification of those at a higher risk of developing PD or those already exhibiting motor symptoms consistent with neurodegeneration. In this way, these individuals could receive medical attention earlier in their disease course to slow or delay disease progression. Furthermore, appropriate post-concussive care could help support a brain damaged by TBI and prevent unnecessary inflammatory-induced trauma in certain patients. Lastly, it is worth mentioning that some people with TBI who do not develop PD have an unknown protective mechanism. If so, understanding this mechanism which protects people with TBI from developing PD may also help with treatment options for people at risk of developing PD.

## Figures and Tables

**Figure 1 ijms-24-07186-f001:**
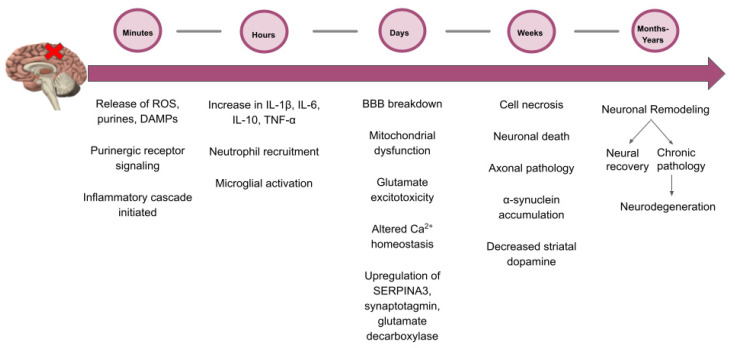
The timeline of inflammatory, immunological, cellular, and neural events occurs minutes to years following TBI.

**Figure 2 ijms-24-07186-f002:**
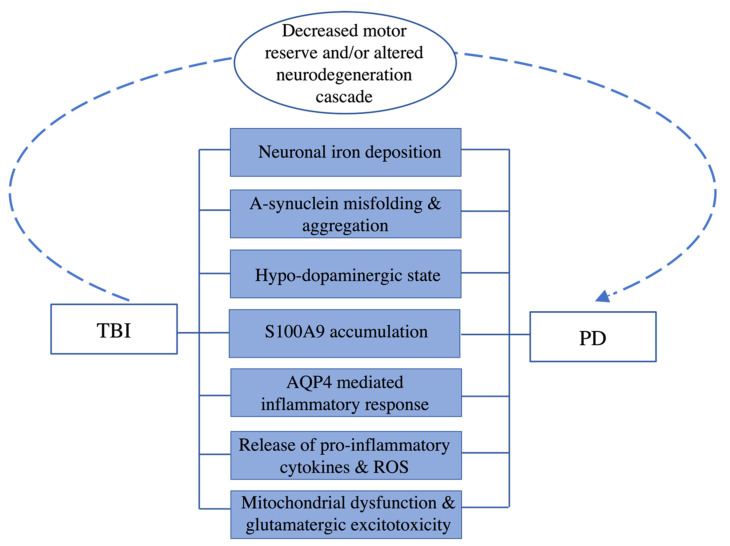
The relationship between TBI and PD, with fundamental pathologic changes of both disorders in the center blue boxes. The dotted line and associated text represent a proposed mechanism by which TBI may induce PD development.

**Table 1 ijms-24-07186-t001:** PD drugs, medicine properties, original references, target, trial phase.

Drug Name	Medical Properties	Original Reference	Target	Trial Phase
Levodopa	Converted to dopamine in the brain	[70]	Dopamine receptor	N/A
Carbidopa- Levodopa	Increases dopamine levels in the brain	[70]	Dopamine receptor	N/A
Dopamine agonists	Mimics the effects of dopamine in the brain	[70]	Dopamine receptor	N/A
MAO-B inhibitor	Blocks the breakdown of dopamine in the brain	[70]	Monoamine oxidase-B	N/A
COMT inhibitor	Blocks the breakdown of dopamine in the brain	[70]	Catechol-O- methyltransferase	N/A
Amantadine	Increases dopamine release and blocks the glutamate receptor	[70]	Dopamine receptor, glutamate receptor	N/A
Apomorphine	Stimulates DAR in the brain	[71]	Dopamine receptor	Phase III
IPX203	Extended-release Carbidopa-levodopa	[72]	Dopamine receptor	Phase III
UCB0599	Activates glucagon-like peptide-1 receptor in the brain	[71]	Glucagon-like peptide-1 receptor	Phase II
PF-06412562	Targets α-Synuclein protein in the brain	[71]	α-Synuclein protein	Phase I
PRX002	Targets α-Synuclein protein in the brain	[71]	α-Synuclein protein	Phase I
CDNF	Neuroprotective protein	[71]	N/A	Phase I
CERE-120	Gene therapy to increase the production of neurotrophic factor	[71]	N/A	Phase I
BIIBO54	Targets α-Synuclein protein in the brain	[71]	α-Synuclein protein	Completed Phase II
Nilotinib	Protein tyrosine kinase inhibitor	[73]	BCR-ABL, c-kit, and PDGF	Phase II
Prasinezumab	a humanized monoclonal antibody	[74]	α-Synuclein protein	Phase II

**Table 2 ijms-24-07186-t002:** Mechanisms of non-pharmacological treatments for PD.

Non-Pharmacological Treatment	Mechanism of Action
Acupuncture	Acupuncture involves inserting fine needles into specific points on the body to stimulate nerves and improve blood flow. In PD, acupuncture has been shown to improve motor symptoms such as tremors, rigidity, and gait difficulties [82].
Hydrotherapy	Hydrotherapy involves exercising in a pool of warm water. The buoyancy of the water reduces the weight and pressure on the joints, making it easier to move. In PD, hydrotherapy has been shown to improve mobility, balance, and muscle strength [82].
Massage therapy	Massage therapy involves manipulating muscles and soft tissues to improve blood flow and relieve tension. In PD, massage therapy has been shown to improve muscle stiffness, reduce anxiety, and improve the overall quality of life [82].
Neuromodulation	Neuromodulation involves using electrical or magnetic stimulation to target specific brain areas and improve motor symptoms. In PD, neuromodulation techniques such as deep brain stimulation (DBS) have been shown to reduce motor symptoms such as tremors and rigidity [83].
Exercise	Exercise involves physical activity to improve cardiovascular health, strength, and flexibility. In PD, exercise has been shown to improve motor symptoms such as balance, gait, and tremors. Exercise can also improve non-motor symptoms such as depression and anxiety [84].

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
