# Peer review of "Pathophysiology and Neuroimmune Interactions Underlying Parkinson’s Disease and Traumatic Brain Injury"

_ijms, 2023, doi:10.3390/ijms24087186_

Round 1

Reviewer 1 Report

Overall, this review discussed the relationship between TBI and PD. I have some comments as follows.

1.     In the second graph, the authors should update the newest TBI data reported by CDC in 2020. (https://www.cdc.gov/TraumaticBrainInjury/data/index.html)

2.     The author needs to explain and highlight this paper's innovative contributions in the introduction.

3.     The abbreviations should be introduced, such as CDC, MHC class II.

4.     It can be better to re-organize or divide the part of “2.4. Microglia, S100A9, and S100B”. As the authors did not clarify the relationship between microglia, S100A9 or S100B in detail.

5.     Also, it is better to move AQP4 part into the iron section. Besides, the author should clarify the mechanism or role of synaptic and cellular plasticity in the process or formation of PD

6.     The authors are recommended to draw a table about the PD drugs, including the medicinal properties, original reference, target, trial phase.

7.     The authors are recommended to clarify the detail or mechanism of Non-pharmacologic treatments.

8.     In the part of “4.1. Cellular and inflammatory changes’, the authors should generalize the key information of AQP4 and AQP mediated pathway. Besides, why the authors highlight the role of AQP in inflammation, but not other receptors.

9.     It can be better to re-organize the part of “4. Neuroimmune and Inflammatory Responses to TBI”, as the two sub-sections are both related to inflammation.

Author Response

Please check the attached cover letter

Reviewer 2 Report

Dear Editor,

The manuscript by Lillian et al. discusses the neuroimmunological interactions and subsequent analogous changes that occur in TBI and PD and and the essential role of aquaporins (particularly AQP4) in its pathology and as a viable therapeutic target,

The work is comprehensive, informative, nicely-written, timely and up-to-date (in most parts). Author was successful in providing some well compiled opinions and summaries.

However, there is a number of major and minor points that would need to be addressed in order to improve the quality of this paper before it can be accepted for publication.

General:
- This work overlooked some essential and up-to-date work regarding the recent advances in AQP target validation and future therapies. I have made some suggestions below but author is encouraged to consider citing updated references throughout the work, whenever possible.

-Abstract: typo- AQP4 not APQ4.

Major:

-The increased AQP expression and the redistribution/surface localization can be two different concepts. For example, previous studies have shown an increased in AQP4 membrane localisation in primary human astrocytes which wasn’t accompanied by a change in AQP4 protein expression levels. This mislocalization can be a potential therapeutic target. References:
https://www.ncbi.nlm.nih.gov/pmc/articles/PMC5765450/
https://www.ncbi.nlm.nih.gov/pubmed/31242419
https://pubmed.ncbi.nlm.nih.gov/26013827/

-The authors omit a key study from Kitchen et al 2020 demonstrating that the development of edema following injury-induced hypoxia is AQP4 dependent. That study shows that CNS edema is associated with increases both in total aquaporin-4 expression and aquaporin-4 subcellular translocation to the blood-spinal-cord-barrier (BSCB). Pharmacological inhibition of AQP-4 translocation to the BSCB prevents the development of CNS edema and promotes functional recovery in injured rats.

This role has been recently been confirmed by the work of Sylvain et al BBA 2021 which has demonstrated that targeting AQP4 effectively reduces cerebral edema during the early acute phase in in stroke using photothrombotic stroke model. They have also shown a link to brain energy metabolism as indicated by the increase of glycogen levels. References to be included:

https://www.cell.com/cell/fulltext/S0092-8674(20)30330-5.

https://pubmed.ncbi.nlm.nih.gov/33561476/

-AQPs have been validated as an important drug target but there is no single drug that has yet been approved to successfully target it, authors need to discuss recent trends in targeting the molecular and signalling mechanisms of AQPs rather than just the traditional approaches. The importance of this new approach has been discussed in these references which should be included to enrich the discussion of current manuscript. References:

https://pubmed.ncbi.nlm.nih.gov/34973181/

https://www.mdpi.com/1422-0067/23/3/1388

Minor:
- Authors are encouraged to briefly discuss future directions following towards the end of their discussion and conclusion. This could include, but not limit to, the use of humanized self-organized models, organoids, 3D cultures and human microvessel-on-a-chip platforms especially those which are amenable for advanced imaging such as TEM and expansion microscopy since they enable real-time monitoring of AQP4 dynamics. References to be included:

https://pubmed.ncbi.nlm.nih.gov/33117784/

https://pubmed.ncbi.nlm.nih.gov/30165870/

Best.

Author Response

Please check the attached cover letter

Reviewer 3 Report

The present manuscript entitles, Pathophysiology and neuroimmune interactions underlying Parkinson’s Disease and traumatic brain injury would be a good contribution to the field. The manuscript is well written. Here the authors focused on PD and TBI relationship and described potential relationships based on several studies. Authors discussed current pharmacologic and non-pharmacologic treatment options along with future possibilities, the etiology and mechanisms underlying PD, and so on.

Pathophysiology of PD, Treatment Modalities, Neuroimmune and Inflammatory Responses to TBI, Relationship Between TBI and PD are well written, informative. I would like to ask the authors to include more animated figures in most of the sections and subsections to make the phenomena easy to understand for all readers. 

Author Response

Please check the attached cover letter

Reviewer 4 Report

[IJMS] Manuscript ID: ijms-2237374, Lillian, A. et al.

Pathophysiology and neuroimmune interactions underlying Parkinson’s

Disease and traumatic brain injury

Traumatic brain injury (TBI) is a risk factor for developing various neurodegenerative diseases, including Parkinson's disease (PD), however, the mechanisms are not much reviewed systematically. In this paper, the authors address overlapping pathologies between TBI and PD, such as accumulation of α-synuclein and neuromelanin aggregates, iron deposition, inflammation, and oxidative stress, suggesting a validity in exploring the relationship between TBI and PD.

I agree that the analysis of the relationship between PD and TBI is important and may be useful for many readers. My comments are as follows.

1.     Given that prion protein (PrP) has been well known for its relevance to sport-related TBI or concussion, it is possible that PrP and α-synuclein, the two amyloidogenic aggregations, might be similarly involved in TBI. I recommend that the authors might discuss this possibility. How about other amyloidogenic aggregations?

2.     Figure 1; time line of TBI pathology is good. Is it possible that therapy intervention?

3.     Figure 2; ‘motor deserve’, which is reminiscent of ‘cognitive reserve’ in Alzheimer's disease is interesting but necessary to explain in the main text.

4.     Since iron accumulation, and oxidative stress by Fenton reaction is described in the text, the word ‘ferroptosis’ may be used in the text and Figure 2.

5.     There are many typos, including different font size, the manuscript should be fixed more carefully.

Author Response

Please check the attached cover letter

Round 2

Reviewer 2 Report

Dear Editor,

The authors have successfully addressed the majority of my comments and concerns in order to improve the quality of the manuscript.

I believe that the new sections, improved ones, and updated references, have contributed to enhancing the clarity of the manuscript, which I can now endorse for publication.

All the best!